# ACTIVE IMAGE INDEXING

**Pierre Fernandez**[1,2,*], **Matthijs Douze**[1], **Hervé Jégou**[1], **Teddy Furon**[2],
[1]Meta AI, FAIR    [2]Centre Inria de l'Université de Rennes
[*]Correspondence: pfz@meta.com.

## ABSTRACT

Image copy detection and retrieval from large databases leverage two components. First, a neural network maps an image to a vector representation, that is relatively robust to various transformations of the image. Second, an efficient but approximate similarity search algorithm trades scalability (size and speed) against quality of the search, thereby introducing a source of error. This paper improves the robustness of image copy detection with *active indexing*, that optimizes the interplay of these two components. We reduce the quantization loss of a given image representation by making imperceptible changes to the image before its release. The loss is back-propagated through the deep neural network back to the image, under perceptual constraints. These modifications make the image more retrievable. Our experiments show that the retrieval and copy detection of activated images is significantly improved. For instance, activation improves by $+40\%$ the Recall1@1 on various image transformations, and for several popular indexing structures based on product quantization and locality sensitivity hashing.

## 1  INTRODUCTION

The traceability of images on a media sharing platform is a challenge: they are widely used, easily edited and disseminated both inside and outside the platform. In this paper, we tackle the corresponding task of Image Copy Detection (ICD), *i.e.* finding whether an image already exists in the database; and if so, give back its identifier. ICD methods power reverse search engines, photography service providers checking copyrights, or media platforms moderating and tracking down malicious content (*e.g.* Microsoft's PhotoDNA (2009) or Apple's NeuralHash (2021)). Image identification systems have to be robust to identify images that are edited (cropping, colorimetric change, JPEG compression . . . ) after their release (Douze et al., 2021; Wang et al., 2022).

The common approach for content-based image retrieval reduces images to high-dimensional vectors, referred to as *representations*. Early representations used for retrieval were hand-crafted features such as color histograms (Swain & Ballard, 1991), GIST (Oliva & Torralba, 2001), or Fisher

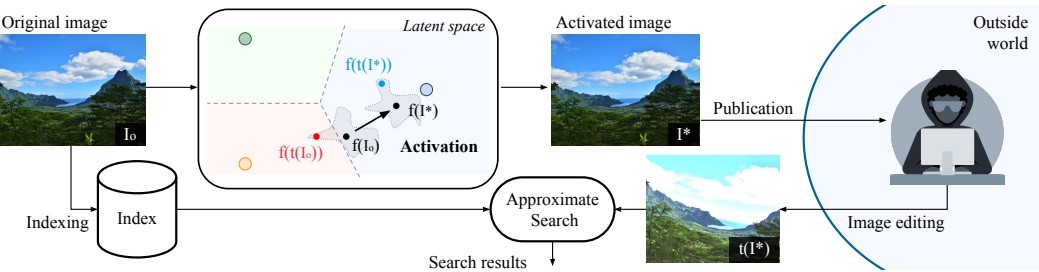

Figure 1: Overview of the method and latent space representation. We start from an original image $I_o$ that can be edited $t(\cdot)$ in various ways: its feature extraction $f(t(I_o))$ spawns the shaded region in the embedding space. The edited versions should be recoverable by nearest neighbor search on quantized representations. In the regular (non-active) case, $f(I_o)$ is quantized by the index as ●. When the image is edited, $t(I_o)$ switches cells and the closest neighbor returned by the index is the wrong one ●. In active indexing: $I_o$ is modified in an imperceptible way to generate $I^\star$ such that $f(I^\star)$ is further away from the boundary. When edited copies $f(t(I^\star))$ are queried, retrieval errors are significantly reduced.

Vectors (Perronnin et al., 2010). As of now, a large body of work on self-supervised learning focuses on producing discriminative representations with deep neural networks, which has inspired recent ICD systems. In fact, *all* submissions to the NeurIPS2021 Image Similarity challenge (Papakipos et al., 2022) exploit neural networks. They are trained to provide invariance to potential image transformations, akin to data augmentation in self-supervised learning.

Scalability is another key requirement of image similarity search: searching must be fast on large-scale databases, which exhaustive vector comparisons cannot do. In practice, ICD engines leverage approximate neighbor search algorithms, that trade search accuracy against scalability. Approximate similarity search algorithms speed up the search by *not* computing the exact distance between all representations in the dataset (Johnson et al., 2019; Guo et al., 2020). First they lower the number of scored items by partitioning the representation space, and evaluate the distances of only a few subsets. Second, they reduce the computational cost of similarity evaluation with quantization or binarization. These mechanisms make indexing methods subject to the curse of dimensionality. In particular, in high-dimensional spaces, vector representations lie close to boundaries of the partition (Böhm et al., 2001). Since edited versions of an original image have noisy vector representations, they sometimes fall into different subsets or are not quantized the same way by the index. All in all, it makes approximate similarity search very sensitive to perturbations of the edited image representations, which causes images to evade detection.

In this paper, we introduce a method that improves similarity search on large databases, provided that the platform or photo provider can modify the images before their release (see Fig. 1). We put the popular saying "attack is the best form of defense" into practice by applying image perturbations and drawing inspiration from adversarial attacks. Indeed, representations produced with neural networks are subject to *adversarial examples* (Szegedy et al., 2013): small perturbations of the input image can lead to very different vector representations, making it possible to create adversarial queries that fool image retrieval systems (Liu et al., 2019; Tolias et al., 2019; Dolhansky & Ferrer, 2020). In contrast, we modify an image to make it *more* indexing friendly. With minimal changes in the image domain, the image representation is pushed towards the center of the indexing partition, rising the odds that edited versions will remain in the same subset. This property is obtained by minimizing an indexation loss by gradient descent back to the image pixels, like for adversarial examples. For indexing structures based on product quantization (Jegou et al., 2010), this strategy amounts to pushing the representation closer to its quantized codeword, in which case the indexation loss is simply measured by the reconstruction error. Since the image quality is an important constraint here, the perturbation is shaped by perceptual filters to remain invisible to the human eye.

Our contributions are:

- a new approach to improve ICD and retrieval, when images can be changed before release;
- an adversarial image optimization scheme that adds minimal perceptual perturbations to images in order to reduce reconstruction errors, and improve vector representation for indexing;
- experimental evidence that the method significantly improves index performance.

## 2 Preliminaries: Representation Learning and Indexing

For the sake of simplicity, the exposure focuses on image representations from SSCD networks (Pizzi et al., 2022) and the indexing technique IVF-PQ (Jegou et al., 2010), since both are typically used for ICD. Extensions to other methods can be found in Sec. 5.4.

### 2.1 Deep descriptor learning

Metric embedding learning aims to learn a mapping $f : \mathbb{R}^{c \times h \times w} \to \mathbb{R}^d$, such that measuring the similarity between images $I$ and $I'$ amounts to computing the distance $\|f(I) - f(I')\|$. In recent works, $f$ is typically a neural network trained with self-supervision on raw data to learn metrically meaningful representations. Methods include contrastive learning (Chen et al., 2020), self-distillation (Grill et al., 2020; Caron et al., 2021), or masking random patches of images (He et al., 2022; Assran et al., 2022). In particular, SSCD (Pizzi et al., 2022) is a training method specialized for ICD. It employs the contrastive self-supervised method SimCLR (Chen et al., 2020) and entropy regularization (Sablayrolles et al., 2019) to improve the distribution of the representations.

## 2.2 INDEXING

Given a dataset $\mathcal{X} = \{x_i\}_{i=1}^n \subset \mathbb{R}^d$ of $d$-dimensional vector representations extracted from $n$ images and a query vector $x_q$, we consider the indexing task that addresses the problem:

$$x^* := \operatorname*{argmin}_{x \in \mathcal{X}} \|x - x_q\|. \tag{1}$$

This exact nearest neighbor search is not tractable over large-scale databases. Approximate search algorithms lower the amount of scored items thanks to space partitioning and/or accelerate the computations of distances thanks to quantization and pre-computation.

**Space partitioning and cell-probe algorithms.** As a first approximation, nearest neighbors are sought only within a fraction of $\mathcal{X}$: at indexing time, $\mathcal{X}$ is partitioned into $\mathcal{X} = \bigcup_{i=1}^b \mathcal{X}_i$. At search time, an algorithm $Q : \mathbb{R}^d \to \{1, .., b\}^{k'}$ determines a subset of $k'$ buckets in which to search, such that $k' = |Q(x_q)| \ll b$, yielding the approximation:

$$\operatorname*{argmin}_{x \in \mathcal{X}} \|x - x_q\| \approx \operatorname*{argmin}_{x \in \bigcup_{i \in Q(x_q)} \mathcal{X}_i} \|x - x_q\|. \tag{2}$$

A well known partition is the KD-tree (Bentley, 1975) that divides the space along predetermined directions. Subsequently, locality sensitive hashing (LSH) (Indyk & Motwani, 1998; Gionis et al., 1999) and derivative (Datar et al., 2004; Paulevé et al., 2010) employ various hash functions for bucket assignment, which implicitly partitions the space.

We focus on the popular clustering and Inverted Files methods (Sivic & Zisserman, 2003), herein denoted by IVF. They employ a codebook $\mathcal{C} = \{c_i\}_{i=1}^k \subset \mathbb{R}^d$ of $k$ centroids (also called "visual words" in a local descriptor context), for instance learned with k-means over a training set of representations. Then, $Q$ associates $x$ to its nearest centroid $q_c(x)$ such that the induced partition is the set of the $k$ Voronoï cells. When indexing $x$, the IVF stores $x$ in the bucket associated with $c_i = q_c(x)$. When querying $x_q$, IVF searches only the $k'$ buckets associated to centroids $c_i$ nearest to $x_q$.

**Efficient metric computation and product quantization.** Another approximation comes from compressed-domain distance estimation. Vector Quantization (VQ) maps a representation $x \in \mathbb{R}^d$ to a codeword $q_f(x) \in \mathcal{C} = \{C_i\}_{i=1}^K$. The function $q_f$ is often referred to a *quantizer* and $C_i$ as a *reproduction value*. The vector $x$ is then stored as an integer in $\{1, .., K\}$ corresponding to $q_f(x)$. The distance between $x$ and query $x_q$ is approximated by $\|q_f(x) - x_q\|$, which is an "asymmetric" distance computation (ADC) because the query is not compressed. This leads to:

$$\operatorname*{argmin}_{x \in \mathcal{X}} \|x - x_q\| \approx \operatorname*{argmin}_{x \in \mathcal{X}} \|q_f(x) - x_q\|. \tag{3}$$

Binary quantizers (a.k.a. sketches, Charikar (2002) lead to efficient computations but inaccurate distance estimates (Weiss et al., 2008). Product Quantization (PQ) (Jegou et al., 2010) or derivatives Ge et al. (2013) offer better estimates. In PQ, a vector $x \in \mathbb{R}^d$ is split into $m$ subvectors in $\mathbb{R}^{d/m}$: $x = (u^1, \ldots, u^m)$. The product quantizer then quantizes the subvectors: $q_f : x \mapsto (q^1(u^1), \ldots, q^m(u^m))$. If each subquantizer $q^j$ has $K_s$ reproduction values, the resulting quantizer $q_f$ has a high $K = (K_s)^m$. The squared distance estimate is decomposed as:

$$\|q_f(x) - x_q\|^2 = \sum_{j=1}^m \|q^j(u^j) - u_q^j\|^2. \tag{4}$$

This is efficient since $x$ is stored by the index as $q_f(x)$ which has $m \log_2 K_s$ bits, and since summands can be precomputed without requiring decompression at search time.

## 3 ACTIVE INDEXING

Active indexing takes as input an image $I_o$, adds the image representation to the index and outputs an activated image $I^\star$ with better traceability properties for the index. It makes the feature representation produced by the neural network more compliant with the indexing structure. The activated image is the one that is disseminated on the platform, therefore the alteration must not degrade the perceived quality of the image.

Images are activated by an optimization on their pixels. The general optimization problem reads:

$$I^{\star} := \underset{I \in \mathcal{C}(I_o)}{\operatorname{argmin}} \ \mathcal{L}\left(I; I_o\right), \tag{5}$$

where $\mathcal{L}$ is an indexation loss dependent on the indexing structure, $\mathcal{C}(I_o)$ is the set of images perceptually close to $I_o$. Algorithm 1 and Figure 1 provide an overview of active indexing.

### 3.1 IMAGE OPTIMIZATION DEDICATED TO IVF-PQ ("ACTIVATION")

The indexing structure IVF-PQ involves a coarse quantizer $q_{\text{c}}$ built with k-means clustering for space partitioning, and a fine product quantizer $q_{\text{f}}$ on the residual vectors, such that a vector $x \in \mathbb{R}^d$ is approximated by $q(x) = q_{\text{c}}(x) + q_{\text{f}}\left(x - q_{\text{c}}(x)\right)$.

We solve the optimization problem (5) by iterative gradient descent, back-propagating through the neural network back to the image. The method is classically used in adversarial example generation (Szegedy et al., 2013; Carlini & Wagner, 2017) and watermarking (Vukotić et al., 2020; Fernandez et al., 2022).

---

**Algorithm 1** Active indexing for IVF-PQ

**Input**: $I_o$: original image; $f$: feature extractor;
Add $x_o = f(I_o)$ to Index, get $q(x_o)$;
Initialize $\delta_0 = 0_{(c \times h \times w)}$;
**for** $t = 0, ..., N - 1$ **do**
    $I_t \leftarrow I_o + \alpha \, . \, H_{\text{JND}}(I_o) \odot \tanh(\delta_t)$
    $x_t \leftarrow f(I_t)$
    $\mathcal{L} \leftarrow \mathcal{L}_{\text{f}}(x_t, q(x_o)) + \lambda \mathcal{L}_{\text{i}}(\delta_t)$
    $\delta_{t+1} \leftarrow \delta_t + \eta \times \text{Adam}(\mathcal{L})$
**end for**
**Output**: $I^{\star} = I_N$ activated image

---

Given an original image $I_o$, the loss is an aggregation of the following objectives:

$$\mathcal{L}_{\text{f}}(x, q(x_o)) = \|x - q(x_o)\|^2 \qquad \text{with } x_o = f(I_o), \, x = f(I) \tag{6}$$

$$\mathcal{L}_{\text{i}}(I, I_o) = \|I - I_o\|^2. \tag{7}$$

$\mathcal{L}_{\text{i}}$ is a regularization on the image distortion. $\mathcal{L}_{\text{f}}$ is the indexation loss that operates on the representation space. $\mathcal{L}_{\text{f}}$ is the Euclidean distance between $x$ and the target $q(x_o)$ and its goal is to push the image feature towards $q(x_o)$. With IVF-PQ as index, the representation of the activated image gets closer to the quantized version of the original representation, but also closer to the coarse centroid. Finally, the losses are combined as $\mathcal{L}(I; I_o) = \mathcal{L}_{\text{f}}(x, q(x_o)) + \lambda \mathcal{L}_{\text{i}}(I, I_o)$.

### 3.2 PERCEPTUAL ATTENUATION

It is common to optimize a perturbation $\delta$ added to the image, rather than the image itself. The adversarial example literature often considers perceptual constraints in the form of an $\ell_p$-norm bound applied on $\delta$ (Madry et al. (2018) use $\|\delta\|_\infty < \varepsilon = 8/255$). Although a smaller $\varepsilon$ makes the perturbation less visible, this constraint is not optimal for the human visual system (HVS), *e.g.* perturbations are more noticeable on flat than on textured areas of the image (see App. A.2).

We employ a handcrafted perceptual attenuation model based on a Just Noticeable Difference (JND) map (Wu et al., 2017), that adjusts the perturbation intensity according to luminance and contrast masking. Given an image $I$, the JND map $H_{\text{JND}}(I) \in \mathbb{R}^{c \times h \times w}$ models the minimum difference perceivable by the HVS at each pixel and additionally rescales the perturbation channel-wise since the human eye is more sensible to red and green than blue color shift (see App. A for details).

The relation that links the image $I$ sent to $f$, $\delta$ being optimized and the original $I_o$, reads:

$$I = I_o + \alpha \, . \, H_{\text{JND}}(I_o) \odot \tanh(\delta), \tag{8}$$

with $\alpha$ a global scaling parameter that controls the strength of the perturbation and $\odot$ the pointwise multiplication. Coupled with the regularization $\mathcal{L}_{\text{i}}$ (6), it enforces that the activated image is perceptually similar, *i.e.* $I^{\star} \in \mathcal{C}(I_o)$ as required in (5).

### 3.3 IMPACT ON THE INDEXING PERFORMANCE

Figure 1 illustrates that the representation of the activated image gets closer to the reproduction value $q(f(I_o))$, and farther away from the Voronoï boundary. This is expected to make image similarity search more robust because (1) it decreases the probability that $x = f(t(I_o))$ "falls" outside the bucket; and (2) it lowers the distance between $x$ and $q(x)$, improving the PQ distance estimate.

Besides, by design, the representation stored by the index is invariant to the activation. Formally stated, consider two images $I$, $J$, and one activated version $J^\star$ together with their representations $x, y, y^\star$. When querying $x = f(I)$, the distance estimate is $\|q(y^\star) - x\| = \|q(y) - x\|$, so the index is oblivious to the change $J \to J^\star$. This means that the structure can index passive and activated images at the same time. Retrieval of activated images is more accurate but the performance on passive images does not change. This compatibility property makes it possible to select only a subset of images to activate, but also to activate already-indexed images at any time.

# 4 ANALYSES

We provide insights on the method for IVF-PQ, considering the effects of quantization and space partitioning. For an image $I$ whose representation is $x = f(I) \in \mathbb{R}^d$, $\hat{x}$ denotes the representation of a transformed version: $\hat{x} = f(t(I)) \in \mathbb{R}^d$, and $x^\star$ the representation of the activated image $I^\star$. For details on the images and the implementation used in the experimental validations, see Sec. 5.1.

## 4.1 PRODUCT QUANTIZATION: IMPACT ON DISTANCE ESTIMATE

We start by analyzing the distance estimate considered by the index:

$$\|\hat{x} - q(x)\|^2 = \|x - q(x)\|^2 + \|\hat{x} - x\|^2 + 2(x - q(x))^\top (\hat{x} - x). \quad (9)$$

The activation aims to reduce the first term, *i.e.* the quantization error $\|x - q(x)\|^2$, which in turn reduces $\|\hat{x} - q(x)\|^2$. Figure 3 shows in blue the empirical distributions of $\|x - q(x)\|^2$ (passive) and $\|x^\star - q(x)\|^2$ (activated). As expected the latter has a lower mean, but also a stronger variance. The variation of the following factors may explain this: *i)* the strength of the perturbation (due to the HVS modeled by $H_{\text{JND}}$ in (8)), *ii)* the sensitivity of the feature extractor $\|\nabla_x f(x)\|$ (some features are easier to push than others), *iii)* the shapes and sizes of the Voronoï cells of PQ.

The second term of (9) models the impact of the image transformation in the feature space. Comparing the orange and blue distributions in Fig. 3, we see that it has a positive mean, but the shift is bigger for activated images. We can assume that the third term has null expectation for two reasons: *i)* the noise $\hat{x} - x$ is independent of $q(x)$ and centered around 0, *ii)* in the high definition regime, quantification noise $x - q(x)$ is independent of $x$ and centered on 0. Thus, this term only increases the variance. Since $x^\star - q(x)$ has smaller norm, this increase is smaller for activated images.

All in all, $\|\hat{x}^\star - q(x)\|^2$ has a lower mean but a stronger variance than its passive counterpart $\|\hat{x} - q(x)\|^2$. Nevertheless, the decrease of the mean is so large that it compensates the larger variance. The orange distribution in active indexing is further away from the green distribution for negative pairs, *i.e.* the distance between an indexed vector $q(x)$ and an independent query $y$.

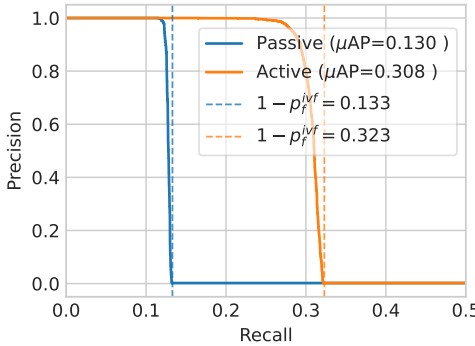

Figure 2: Precision-Recall curve for ICD with 50k queries and 1M reference images (more details for the experimental setup in Sec. 5.1). $p_f^{\text{ivf}}$ is the probability of failure of the IVF (Sec. 4.2).

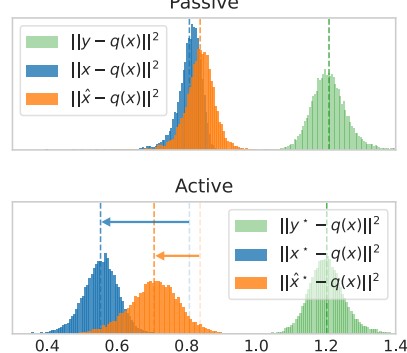

Figure 3: Distance estimates histograms (sec. 4.1). With active indexing, $\|x - q(x)\|^2$ is reduced ($\leftarrow$), inducing a shift ($\leftarrow$) in the distribution of $\|\hat{x} - q(x)\|^2$, where $t(I)$ a hue-shifted version of $I$. $y$ is a random query.

## 4.2 SPACE PARTITIONING: IMPACT ON THE IVF PROBABILITY OF FAILURE

We denote by $p_f := \mathbb{P}(q_c(x) \neq q_c(\hat{x}))$ the probability that $\hat{x}$ is assigned to a wrong bucket by IVF assignment $q_c$. In the single-probe search ($k' = 1$), the recall (probability that a pair is detected when it is a true match, for a given threshold $\tau$ on the distance) is upper-bounded by $1 - p_f$:

$$R_\tau = \mathbb{P}\left(\{q_c(\hat{x}) = q_c(x)\} \cap \{\|\hat{x} - q(x)\| < \tau\}\right) \leq \mathbb{P}\left(\{q_c(\hat{x}) = q_c(x)\}\right) = 1 - p_f. \quad (10)$$

In other terms, even with a high threshold $\tau \to \infty$ (and low precision), the detection misses representations that ought to be matched, with probability $p_f$. It explains the sharp drop at recall $R = 0.13$ in Fig. 2. This is why it is crucial to decrease $p_f$. The effect of active indexing is to reduce $\|\hat{x} - q_c(x)\|$ therefore reducing $p_f$ and increasing the upper-bound for $R$: the drop shifts towards $R = 0.32$.

This explanation suggests that pushing $x$ towards $q_c(x)$ decreases even more efficiently $p_f$. This makes the IVF more robust to transformation but this may jeopardize the PQ search because features of activated images are packed altogether. In a way, our strategy, which pushes $x$ towards $q(x)$, dispatches the improvement over the IVF and the PQ search.

## 5 EXPERIMENTAL RESULTS

### 5.1 EXPERIMENTAL SETUP

**Dataset.**   We use DISC21 (Douze et al., 2021) a dataset dedicated to ICD. It includes 1M reference images and 50k query images, 10k of which are true copies from reference images. A disjoint 1M-image set with same distribution as the reference images is given for training. Images resolutions range from 200×200 to 1024×1024 pixels (most of the images are around 1024×768 pixels).

The queries used in our experiments are *not* the queries in DISC21, since we need to control the image transformations in our experiments, and most transformations of DISC21 were done manually so they are not reproducible. Our queries are transformations of images *after active indexing*. These transformations range from simple attacks like rotation to more realistic social network transformations which created the original DISC21 queries (see App. B.1).

**Metrics.**   For retrieval, our main metric is Recall $1@1$ ($R@1$ for simplicity), which corresponds to the proportion of positive queries where the top-1 retrieved result is the reference image.

For copy detection, we use the same metric as the NeurIPS Image Similarity Challenge (Douze et al., 2021). We retrieve the $k = 10$ most similar database features for every query; and we declare a pair is a match if the distance is lower than a threshold $\tau$. To evaluate detection efficiency, we use the 10k matching queries above-mentioned together with 40k negative queries (*i.e.* not included in the database). We use precision and recall, as well as the area under the precision-recall curve, which is equivalent to the micro average precision ($\mu AP$). While $R@1$ only measures ranking quality of the index, $\mu AP$ takes into account the confidence of a match.

As for image quality metric, we use the Peak Signal-to-Noise Ratio (PSNR) which is defined as $10 \log_{10}\left(255^2/\mathrm{MSE}(I, I')^2\right)$, as well as SSIM (Wang et al., 2004) and the norm $\|I - I'\|_\infty$.

**Implementation details.**   The evaluation procedure is: (1) we train an index on the 1M training images, (2) index the 1M reference images, (3) activate (or not) 10k images from this reference set. (4) At search time, we use the index to get closest neighbors (and their distances) of transformed versions from a query set made of the 10k images.

Unless stated otherwise, we use a IVF4096,PQ8x8 index (IVF quantizer with 4096 centroids, and PQ with 8 subquantizers of $2^8$ centroids), and use only one probe on IVF search for shortlist selection ($k' = 1$). Compared to a realistic setting, we voluntarily use an indexing method that severely degrades learned representations to showcase and analyze the effect of the active indexing. For feature extraction, we use an SSCD model with a ResNet50 trunk (He et al., 2016). It takes image resized to 288×288 and generates normalized representations in $\mathbb{R}^{512}$. Optimization (5) is done with the Adam optimizer (Kingma & Ba, 2015), the learning rate is set to 1, the number of iterations to $N = 10$ and the regularization to $\lambda = 1$. In (8), the distortion scaling is set to $\alpha = 3$ (leading to an average PNSR around $43$ dB). In this setup, activating 128 images takes around 6s ($\approx$ 40ms/image) with a 32GB GPU. It can be sped-up at the cost of some accuracy (see App. C.2).

Table 1: Comparison of the index performance between activated and passive images. The search is done on a 1M image set and $R@1$ is averaged over 10k query images submitted to different transformations before search. **Random**: randomly apply 1 to 4 transformations. **Avg.**: average on the transformations presented in the table (details in App. B.2). **No index**: exhaustive brute-force nearest neighbor search. **IVF-PQ**: IVF4096,PQ8x8 index with $k'$=1 (16 for **IVF-PQ**[16]). **IVF-PQ**[†]: IVF512,PQ32x8 with $k' = 32$.

| | Search (ms) | Bytes/vector | Activated | Identity | Contr. 0.5 | Contr. 2.0 | Bright. 0.5 | Bright. 2.0 | Hue 0.2 | Blur 2.0 | JPEG 50 | Rot. 25 | Rot. 90 | Crop 0.5 | Resi. 0.5 | Meme | Random | Avg. |
|---|---|---|---|---|---|---|---|---|---|---|---|---|---|---|---|---|---|---|
| No index | 252 | 2048 | ✗ | 1.00 | 1.00 | 1.00 | 1.00 | 1.00 | 1.00 | 1.00 | 1.00 | 1.00 | 1.00 | 1.00 | 1.00 | 1.00 | 0.90 | 0.99 |
| IVF-PQ | 0.38 | 8 | ✗ | 1.00 | 0.73 | 0.39 | 0.73 | 0.28 | 0.62 | 0.48 | 0.72 | 0.07 | 0.14 | 0.14 | 0.72 | 0.14 | 0.13 | 0.45 |
| | | | ✔ | 1.00 | 1.00 | 0.96 | 1.00 | 0.92 | 1.00 | 0.96 | 0.99 | 0.10 | 0.50 | 0.29 | 1.00 | 0.43 | 0.32 | 0.75 |
| IVF-PQ[16] | 0.42 | 8 | ✗ | 1.00 | 1.00 | 0.90 | 1.00 | 0.78 | 0.99 | 0.95 | 0.99 | 0.35 | 0.57 | 0.57 | 1.00 | 0.56 | 0.39 | 0.79 |
| | | | ✔ | 1.00 | 1.00 | 1.00 | 1.00 | 0.98 | 1.00 | 1.00 | 1.00 | 0.43 | 0.88 | 0.75 | 1.00 | 0.84 | 0.50 | 0.88 |
| IVF-PQ[†] | 1.9 | 32 | ✗ | 1.00 | 1.00 | 0.99 | 1.00 | 0.95 | 1.00 | 0.99 | 1.00 | 0.72 | 0.87 | 0.88 | 1.00 | 0.87 | 0.61 | 0.92 |
| | | | ✔ | 1.00 | 1.00 | 0.99 | 1.00 | 0.98 | 1.00 | 1.00 | 1.00 | 0.75 | 0.92 | 0.91 | 1.00 | 0.92 | 0.63 | 0.94 |

## 5.2 ACTIVE VS. PASSIVE

This section compares retrieval performance of active and passive indexing. We evaluate $R@1$ when different transformations are applied to the 10k reference images before search. The "Passive" lines of Tab. 1 show how the IVF-PQ degrades the recall. This is expected, but the IVF-PQ also accelerates search $500\times$ and the index is $256\times$ more compact, which is necessary for large-scale applications. Edited images are retrieved more often when they were activated for the index: increase of up to $+60$ $R@1$ for strong brightness and contrast changes, close to results of the brute-force search. We also notice that the performance of the active IVF-PQ$^{k'=1}$ is approximately the same as the one of the passive IVF-PQ$^{k'=16}$, meaning that the search can be made more efficient at equal performance. For the IVF-PQ$^{†}$ that does less approximation in the search (but is slower and takes more memory), retrieval on activated images is also improved, though to a lesser extent.

As for copy detection, Figure 2 gives the precision-recall curves obtained for a sliding value of $\tau$, and corresponding $\mu$AP. Again, we observe a significant increase ($\times 2$) in $\mu$AP with active indexing. Note that the detection performance is much weaker than the brute-force search even in the active case because of the strong approximation made by space partitioning (more details in Sec. 4.2).

Example of activated images are given in Fig. 5 (more in App. E), while the qualitative image metrics are as follows: PSNR$= 43.8 \pm 2.2$ dB, SSIM$= 0.98 \pm 0.01$, and $\|I - I'\|_\infty = 14.5 \pm 1.2$. These results are computed on 10k images, the $\pm$ indicates the standard deviation.

## 5.3 IMAGE QUALITY TRADE-OFF

For a fixed index and neural extractor, the performance of active indexing mainly depends on the scaling $\alpha$ that controls the activated image quality. In Fig. 4, we repeat the previous experiment for different values of $\alpha$ and plot the $\mu$AP against the average PSNR. As expected, lower PSNR implies better $\mu$AP. For instance, at PSNR 30 dB, the $\mu$AP is augmented threefold compared to the passive case. Indeed, for strong perturbations the objective function of (6) can be further lowered, reducing even more the gap between representations and their quantized counterparts.

## 5.4 GENERALIZATION

**Generalization to other neural feature extractors.**

We first reproduce the experiment of Sec. 5.1 with different extractors, that cover distinct training methods and architectures. Among them, we evaluate a ResNext101 (Xie et al., 2017) trained with SSCD (Pizzi et al., 2022), a larger network than the ResNet50 used in our main experiments ; the winner of the descriptor track of the NeurIPS ISC, LYAKAAP-dt1 (Yokoo, 2021), that uses an EfficientNetv2 architecture (Tan & Le, 2021) ; networks from DINO (Caron et al., 2021), either based on ResNet50 or ViT (Dosovitskiy et al., 2021), like the ViT-S model (Touvron et al., 2021).

Table 2: $R@1$ for different transformations before search. We use our method to activate images for indexing with IVF-PQ, with different neural networks used as feature extractors.

| Name | Architecture | Activated | Identity | Contr. 0.5 | Contr. 2.0 | Bright. 0.5 | Bright. 2.0 | Hue 0.2 | Blur 2.0 | JPEG 50 | Rot. 25 | Rot. 90 | Crop 0.5 | Resi. 0.5 | Meme | Random | Avg. |
|------|--------------|-----------|----------|------------|------------|-------------|-------------|---------|----------|---------|---------|---------|----------|-----------|------|--------|------|
| SSCD | ResNet50 | ✗ | 1.00 | 0.73 | 0.39 | 0.73 | 0.28 | 0.62 | 0.48 | 0.72 | 0.07 | 0.14 | 0.14 | 0.72 | 0.14 | 0.13 | 0.45 |
| | | ✔ | 1.00 | 1.00 | 0.96 | 1.00 | 0.92 | 1.00 | 0.96 | 0.99 | 0.10 | 0.50 | 0.29 | 1.00 | 0.43 | 0.32 | 0.75 |
| | ResNext101 | ✗ | 1.00 | 0.88 | 0.68 | 0.88 | 0.57 | 0.84 | 0.46 | 0.79 | 0.46 | 0.63 | 0.53 | 0.80 | 0.48 | 0.28 | 0.66 |
| | | ✔ | 1.00 | 1.00 | 0.96 | 1.00 | 0.90 | 0.99 | 0.77 | 0.97 | 0.53 | 0.85 | 0.64 | 1.00 | 0.74 | 0.37 | 0.84 |
| DINO | ResNet50 | ✗ | 1.00 | 0.66 | 0.65 | 0.65 | 0.52 | 0.71 | 0.52 | 0.82 | 0.07 | 0.20 | 0.51 | 0.84 | 0.62 | 0.18 | 0.57 |
| | | ✔ | 1.00 | 0.99 | 0.88 | 0.99 | 0.75 | 0.93 | 0.72 | 0.94 | 0.08 | 0.25 | 0.57 | 0.99 | 0.82 | 0.23 | 0.72 |
| | ViT-s | ✗ | 1.00 | 0.89 | 0.71 | 0.86 | 0.64 | 0.75 | 0.74 | 0.90 | 0.14 | 0.18 | 0.57 | 0.88 | 0.61 | 0.25 | 0.65 |
| | | ✔ | 1.00 | 0.99 | 0.94 | 0.99 | 0.92 | 0.98 | 0.89 | 0.99 | 0.15 | 0.28 | 0.63 | 0.99 | 0.77 | 0.32 | 0.77 |
| ISC-dt1 | EffNetv2 | ✗ | 1.00 | 0.25 | 0.08 | 0.16 | 0.01 | 0.51 | 0.54 | 0.84 | 0.18 | 0.16 | 0.23 | 0.79 | 0.16 | 0.18 | 0.36 |
| | | ✔ | 1.00 | 0.57 | 0.16 | 0.33 | 0.01 | 0.88 | 0.79 | 0.97 | 0.20 | 0.24 | 0.29 | 0.97 | 0.26 | 0.26 | 0.49 |

Table 2 presents the $R@1$ obtained on 10k activated images when applying different transformations before search. The $R@1$ is better for activated images for all transformations and all neural networks. The average improvement on all transformations ranges from $+12\%$ for DINO ViT-s to $+30\%$ for SSCD ResNet50.

**Generalization to other indexes.**

The method easily generalizes to other types of indexing structures, the only difference being in the indexation loss $\mathcal{L}_\mathrm{f}$ (6). We present some of them below:

- **PQ and OPQ**. In PQ (Jegou et al., 2010), a vector $x \in \mathbb{R}^d$ is approximated by $q_\mathrm{f}(x)$. $\mathcal{L}_\mathrm{f}$ reads $\|x - q_\mathrm{f}(x_o)\|$. In OPQ (Ge et al., 2013), vectors are rotated by matrix $R$ before codeword assignment, such that $RR^\top = I$. $\mathcal{L}_\mathrm{f}$ becomes $\|x - R^\top q_\mathrm{f}(Rx_o)\|$.

- **IVF.** Here, we only do space partitioning. Employing $\mathcal{L}_\mathrm{f} = \|x - q_\mathrm{c}(x_o)\|$ ("pushing towards the cluster centroid") decreases the odds of $x$ falling in the wrong cell (see Sec. 4.2). In this case, an issue can be that similar representations are all pushed together to a same centroid, which makes them less discriminate. Empirically, we found that this does not happen because perceptual constraint in the image domain prevents features from getting too close.

- **LSH.** Locality Sensitive Hashing maps $x \in \mathbb{R}^d$ to a binary hash $b(x) \in \mathbb{R}^L$. It is commonly done with projections against a set of vectors, which give for $j \in [1, .., L]$, $b_j(x) = \mathrm{sign}(w_j^\top x)$. The objective $\mathcal{L}_\mathrm{f} = -1/L \sum_j \mathrm{sign}(b(x_o)) \cdot w_j^\top x$, allows to push $x$ along the LSH directions and to improve the robustness of the hash.

Table 3 presents the $R@1$ and $\mu$AP obtained on the 50k query set. Again, results are always better in the active scenario. We remark that active indexing has more impact on space partitioning tech-

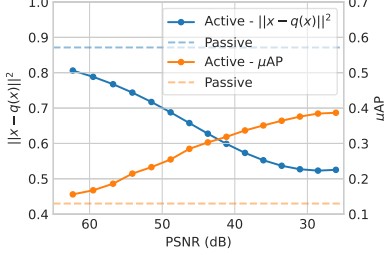

Figure 4: PSNR trade-off. As the PSNR decreases, the μAP (orange) gets better, because the distance (blue) between activated representations $x$ and $q(x)$ decreases.

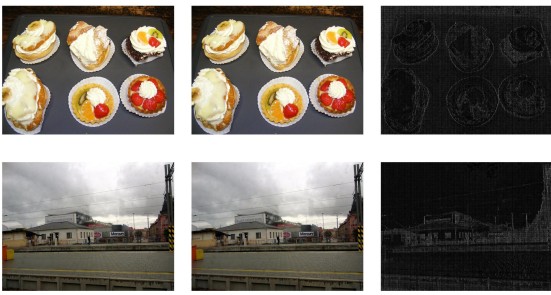

Figure 5: Activated images. *Left:* reference from DISC (R000643.jpg and R000761.jpg), *middle:* activated image, right: pixel-wise difference.

Table 3: $R@1$ averaged on transformations presented in Tab. 1 and $\mu$AP for different indexing structures

| Index | Search time | $R@1$ avg. | | $\mu$AP | |
|---|---|---|---|---|---|
| | | Passive | Activated | Passive | Activated |
| IVF 1024 | 0.32 ms | 0.66 | **0.89** | 0.16 | **0.43** |
| PCA64, LSH | 0.99 ms | 0.78 | **0.86** | 0.25 | **0.39** |
| OPQ 8x8 | 5.71 ms | 0.92 | **0.94** | 0.48 | **0.55** |

niques: the improvement for IVF is higher than with PQ and the LSH binary sketches. As to be expected, the impact is smaller when the indexing method is more accurate.

## 6 RELATED WORK

**Image watermarking** hides a message in a host image, such that it can be reliably decoded even if the host image is edited. Early methods directly embed the watermark signal in the spatial or transform domain like DCT or DWT (Cox et al., 2007). Recently, deep-learning based methods jointly train an encoder and a decoder to learn how to watermark images (Zhu et al., 2018; Ahmadi et al., 2020; Zhang et al., 2020).

Watermarking is an alternative technology for ICD. Our method bridges indexing and watermarking, where the image is modified before publication. Regarding retrieval performance, active indexing is more robust than watermarking. Indeed, the embedded signal reinforces the structure naturally present in the original image, whereas watermarking has to hide a large secret keyed signal independent of the original feature. App. D provides a more thorough discussion and experimental results comparing indexing and watermarking.

**Active fingerprint** is more related to our work. As far as we know, this concept was invented by Voloshynovskiy et al. (2012). They consider that the image $I \in \mathbb{R}^N$ is mapped to $x \in \mathbb{R}^N$ by an invertible transform $W$ such that $WW^\top$. The binary fingerprint is obtained by taking the sign of the projections of $x$ against a set of vectors $b_1, ., b_L \in \mathbb{R}^N$ (à la LSH). Then, they change $x$ to strengthen the amplitude of these projections so that their signs become more robust to noise. They recover $I^\star$ with $W^\top$. This scheme is applied to image patches in (Kostadinov et al., 2016) where the performance is measured as a bit error rate after JPEG compression. Our paper adapts this idea from fingerprinting to indexing, with modern deep learning representations and state-of-the-art indexing techniques. The range of transformations is also much broader and includes geometric transforms.

## 7 CONCLUSION & DISCUSSION

We introduce a way to improve ICD in large-scale settings, when images can be changed before release. It leverages an optimization scheme, similar to adversarial examples, that modifies images so that (1) their representations are better suited for indexing, (2) the perturbation is invisible to the human eye. We provide grounded analyses on the method and show that it significantly improves retrieval performance of activated images, on a number of neural extractors and indexing structures.

Activating images takes time (in the order of 10 ms/image) but one advantage is that the database may contain both active and passive images: active indexing does not spoil the performance of passive indexing and vice-versa. This is good for legacy compliance and also opens the door to flexible digital asset management strategies (actively indexing images of particular importance).

The main limitation of the method is that images need to be activated before release. In the case of existing databases where images have already been released, images could still be activated for future releases (meaning that there would be 2 versions of the image online, a passive one that can be retrieved as long as it is not transformed too strongly, and an activated one with better copy detection properties). Another one is that it is not agnostic to the indexing structure and extractor that are used by the similarity search. Finally, an adversary could still break the indexing system in several ways. In a black-box setting, adversarial purification (Shi et al., 2021) could get rid of the perturbation that activated the image. In a semi-white-box setting (knowledge of the feature extractor), targeted mismatch attacks against passive indexing like Tolias et al. (2019) may also degrade the retrieval.

ACKNOWLEDGMENTS

Teddy Furon thanks ANR-AID for funding Chaire SAIDA ANR-20-CHIA-0011-01.

ETHICS STATEMENT

**Societal impact statement.** Content tracing is a double-edged sword. On the one hand, it allows media platforms to more accurately track malicious content (pornographic, terrorist, violent images, *e.g.* Apple's NeuralHash and Microsoft's PhotoDNA) and to protect copyright (*e.g.* Youtube's Content ID). On the other hand it can be used as a means of societal and political censorship, to restrict free speech of specific communities. However, we still believe that research needs to be advanced to improve global moderation in the internet. We also believe that advantages that a better copy detection could bring are more numerous than its drawbacks.

**Environmental impact statement.** We roughly estimated that the total GPU-days used for running all our experiments to 200, or $\approx 5000$ GPU-hours. Experiments were conducted using a private infrastructure and we estimate total emissions to be in the order of a ton $CO_2$eq. Estimations were conducted using the MachineLearning Impact calculator presented in Lacoste et al. (2019). We do not consider in this approximation: memory storage, CPU-hours, production cost of GPUs/ CPUs, etc. as well as the environmental cost of training the neural networks used as feature extractors. Although the cost of the experiments and the method is high, it could possibly allow a reduction of the computations needed in large data-centers thanks to improved performance of indexing structures.

REPRODUCIBILITY STATEMENT

The implementation is available at github.com/facebookresearch/active_indexing. Models used for feature extraction (SSCD, DINO, ISC-dt1) can be downloaded in their respective repositories. It builds upon the open-source Pytorch (Paszke et al., 2019) and FAISS (Johnson et al., 2019) libraries.

The main dataset used in the experiments (DISC21) can be freely downloaded on its webpage https://ai.facebook.com/datasets/disc21-dataset/. Dataset processing is described in App. B.1.

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

# Supplementary material - **Active Image Indexing**

## A  DETAILS ON THE PERCEPTUAL ATTENUATION MODEL

### A.1  JUST NOTICEABLE DIFFERENCE MAP

The maximum change that the human visual system (HVS) cannot perceive is sometimes referred to as the just noticeable difference (JND) Krueger (1989). It is used in many applications, such as image/video watermarking, compression, quality assessment (JND is also used in audio).

JND models in pixel domain directly calculate the JND at each pixel location (*i.e.* how much pixel difference is perceivable by the HVS). The JND map that we use is based on the work of Chou & Li (1995). We use this model for its simplicity, its efficiency and its good qualitative results. More complex HVS models could also be used if even higher imperceptibility is needed (Watson (1993); Yang et al. (2005); Zhang et al. (2008); Jiang et al. (2022) to cite a few). The JND map takes into account two characteristics of the HVS, namely the luminance adaptation (LA) and the contrast masking (CM) phenomena. We follow the same notations as Wu et al. (2017).

The CM map $\mathcal{M}_C$ is a function of the image gradient magnitude $\mathcal{C}_l$ (the Sobel filter of the image):

$$\mathcal{M}_C(x) = 0.115 \times \frac{\alpha \cdot \mathcal{C}_l(x)^{2.4}}{\mathcal{C}_l(x)^2 + \beta^2} \quad \text{, with } \mathcal{C}_l = \sqrt{\nabla_x I(x)^2 + \nabla_y I(x)^2}, \tag{11}$$

where $x$ is the pixel location, $I(x)$ the image intensity, $\alpha = 16$, and $\beta = 26$. It is an increasing function of $\mathcal{C}_l$, meaning that the stronger the gradient is at $x$, the more the image is masking a local perturbation, and the higher the noticeable pixel difference is.

LA takes into account the fact that the HVS presents different sensitivity to background luminance (*e.g.* it is less sensible in dark backgrounds). It is modeled as:

$$\mathcal{L}_A(x) = \begin{cases} 17 \times \left(1 - \sqrt{\dfrac{B(x)}{127}}\right) & \text{if } B(x) < 127 \\ \dfrac{3 \times (B(x) - 127)}{128} + 3 & \text{if } B(x) \geq 127, \end{cases} \tag{12}$$

where $B(x)$ is the background luminance, which is calculated as the mean luminance value of a local patch centered on $x$.

Finally, both effects are combined with a nonlinear additivity model:

$$H_{\text{JND}} = \mathcal{L}_A + \mathcal{M}_C - C \cdot \min\{\mathcal{L}_A, \mathcal{M}_C\}, \tag{13}$$

where $C$ is set to $0.3$ and determines the overlapping effect. For color images, the final RGB heatmap is $H_{\text{JND}} = [\alpha_R H, \alpha_G H, \alpha_B H]$, where $(\alpha_R, \alpha_G, \alpha_B)$ are inversely proportional to the mixing coefficients for the luminance: $(\alpha_R, \alpha_G, \alpha_B) = 0.072/(0.299, 0.587, 0.114)$.

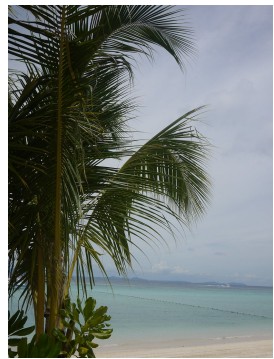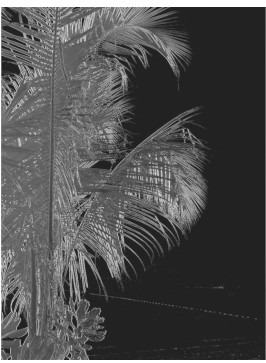

Figure 6: A reference image $I$ from DISC21 (R002815.jpg), and the associated perceptual heatmap $H_{\text{JND}}(I)$.

## A.2   COMPARISON WITH $\ell_\infty$ CONSTRAINT EMBEDDING

Figure 7 shows the same image activated using either the $\ell_\infty$ constraint (commonly used in the adversarial attack literature) or our perceptual constraint based on the JND model explained above. Even with very small $\varepsilon$ (4 over 255 in the example bellow), the perturbation is visible especially in the flat regions of the images, such as the sea or sky.

Laidlaw et al. (2021) also show that the $\ell_\infty$ is not a good perceptual constraint. They use the LPIPS loss (Zhang et al., 2018) as a surrogate for the HVS to develop more imperceptible adversarial attacks. Although a similar approach could be used here, we found that at this small level of image distortion the LPIPS did not capture CM and LA as well as the handcrafted perceptual models present in the compression and watermarking literature.

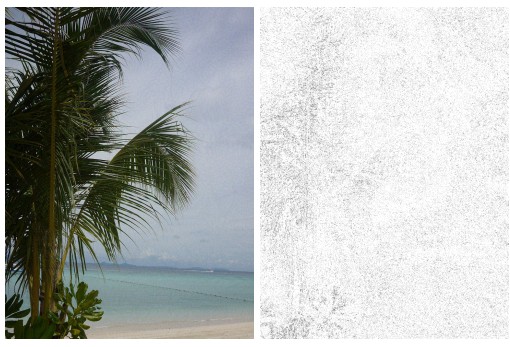 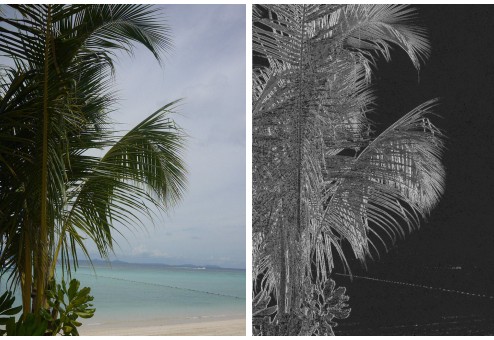

(a) $\ell_\infty = 4$, PSNR $= 36.4$ dB, SSIM $= 0.91$      (b) $\ell_\infty = 23$, PSNR $= 34.4$ dB, SSIM $= 0.94$

Figure 7: Activated images, either with (a) the $\ell_\infty \leq 4$ constraint or with (b) our perceptual model (best viewed on screen). We give the corresponding measures between the original and the protected image, as well as the pixel-wise difference. The perturbation on the right is much less perceptible thanks to the perceptual model, even though its $\ell_\infty$ distance with the original image is much higher.

## B   EXPERIMENTS DETAILS

This section describes the details omitted in experimental sections.

### B.1   DATASET

The dataset DISC 2021 was designed for the Image Similarity Challenge (Douze et al., 2021) and can be downloaded in the dataset webpage: https://ai.facebook.com/datasets/disc21-dataset/.

We want to test performance on edited versions of activated images but in DISC query set transformations are already applied to images. Therefore the query set cannot be used as it is.

We create a first test set "Ref10k" by selecting the 10k images from the reference set that were originally used to generate the queries (the "dev queries" from the downloadable version). We also re-create a query set "Query50k". To be as close as possible, we use the same images that were used for generating queries in DISC. Edited images are generated using the AugLy library (Papakipos & Bitton, 2022), following the guidelines given in the "Automatic Transformations" section of the DISC paper. Therefore, the main difference between the query set used in our experiments and the original one is that ours do not have manual augmentations.

### B.2   TRANSFORMATIONS SEEN AT TEST TIME

They cover both spatial transformations (crops, rotation, etc.), pixel-value transformations (contrast, hue, jpeg, etc.) and "everyday life" transformations with the AugLy augmentations. All transformations are illustrated in Fig. 4. The parameters for all transformations are the ones of the torchvision library (Marcel & Rodriguez, 2010), except for the crop and resize that represent area ratios. For the Gaussian blur transformation we use alternatively $\sigma$, the scaling factor in the exponential, or the kernel size $k_b$ (in torchvision $k_b = (\sigma - 0.35)/0.15$). The "Random" transformation is the one used to

Table 4: Illustration of all transformations evaluated in Tab. 1.

| Identity | Contrast 0.5 | Contrast 2.0 | Brightness 0.5 | Brightness 2.0 |
| --- | --- | --- | --- | --- |

| Hue 0.2 | Blur 2.0 | JPEG 50 | Rotation 25 | Rotation 90 |
| --- | --- | --- | --- | --- |

| Crop 0.5 | Resize 0.5 | Meme | Random |
| --- | --- | --- | --- |

develop the 50k query set. A series of simple 1-4 AugLy transformations are picked at random, with skewed probability for a higher number. Among the possible transformations, there are pixel-level, geometric ones, as well as embedding the image as a screenshot of a social network GUI.

## C   MORE EXPERIMENTAL RESULTS

### C.1   DETAILED METRICS ON DIFFERENT IMAGE TRANSFORMATIONS

On Fig. 8, we evaluate the average $R@1$ over the 10k images from the reference dataset. The experimental setup is the same as for Tab. 1 but a higher number of transformation parameters

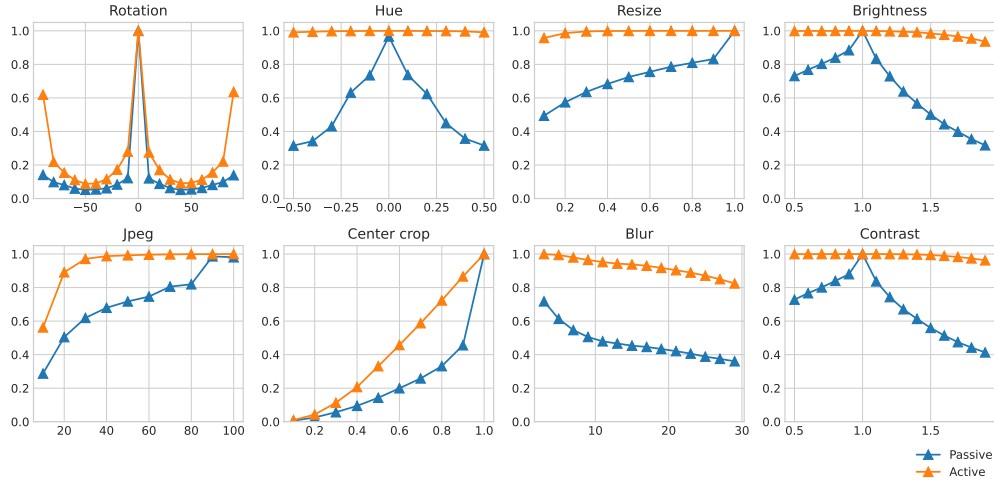

Figure 8: Average $R@1$ comparison between active and passive indexing with IVF-PQ.

are evaluated. As expected, the higher the strength of the transformation, the lower the retrieval performance is. The decrease in performance is significantly reduced with activated images.

## C.2 Additional ablations

**Speeding-up the optimization.** In our experiments, the optimization is done using 10 iterations of gradient descent, which takes approximately 40ms/image. If the indexation time is important (often, this is not the case and only the search time is), it can be reduced at the cost of some accuracy.

We activated 10k reference images, with the same IVF-PQ indexed presented in Sec. 5.2 with only one step of gradient descent with a higher learning rate. Activation times are computed on average. The $R@1$ results in Tab. 5 indicate that the speed-up in the image optimization has a small cost in retrieval accuracy. Specifically, it reduces the $R@1$ for unedited images. The reason is that the learning rate is too high: it can cause the representation to be pushed too far and to leave the indexing cell. This is why a higher number number of steps and a lower learning rate are used in practice. If activation time is a bottleneck, it can however be useful to use less optimization steps.

Table 5: $R@1$ for different transformations applied before search, with either 1 step at learning rate 10, or 10 steps at learning rate 1. Results are averaged on 10k images.

| | Activation | Identity | Contr. 0.5 | Contr. 2.0 | Bright. 0.5 | Bright. 2.0 | Hue 0.2 | Blur 2.0 | JPEG 50 | Rot. 25 | Rot. 90 | Crop 0.5 | Resi. 0.5 | Meme | Random | Avg. |
|---|---|---|---|---|---|---|---|---|---|---|---|---|---|---|---|---|
| Passive | - | 1.00 | 0.73 | 0.39 | 0.73 | 0.28 | 0.62 | 0.48 | 0.72 | 0.07 | 0.14 | 0.14 | 0.72 | 0.14 | 0.13 | 0.45 |
| Adam,lr=1 - 10 steps | 39.8 ms/img | 1.00 | 1.00 | 0.96 | 1.00 | 0.92 | 1.00 | 0.96 | 0.99 | 0.10 | 0.50 | 0.29 | 1.00 | 0.43 | 0.32 | 0.75 |
| Adam,lr=10 - 1 step | 4.3 ms/img | 0.99 | 0.99 | 0.92 | 0.99 | 0.84 | 0.99 | 0.95 | 0.99 | 0.10 | 0.39 | 0.25 | 0.99 | 0.36 | 0.27 | 0.72 |

**Data augmentation at indexing time and EoT.** Expectation over Transformations (Athalye et al., 2018) was originally designed to create adversarial attacks robust to a set of image transformations. We follow a similar approach to improve robustness of the marked image against a set of augmentations $\mathcal{T}$. At each optimization step, we randomly sample $A$ augmentations $\{t_i\}_{i=1}^A$ in $\mathcal{T}$ and consider the average loss: $\mathcal{L}_f = \sum_{i=1}^A \mathcal{L}(I, t_i; I_o)/A$. In our experiments, $\mathcal{T}$ encompasses rotations, Gaussian blurs, color jitters and a differentiable approximation of the JPEG compression Shin & Song (2017). $A$ is set to 8 and we always take the un-augmented image in the chosen set of augmentations.

We activated 10k reference images, with the same IVF-PQ as Sec. 5.2 with or without using EoT. Table 6 shows the average $R@1$ performance over the images submitted to different transformations before search. EoT brings a small improvement, specifically on transformations where base performance is low (*e.g.* rotation or crops here). However, it comes at a higher computational cost since each gradient descent iteration needs $A$ passes through the network, and since fewer images can be jointly activated due to GPU memory limitations (we need to store and back-propagate through $A$ transformations for every image). If the time needed to index or activate an image is not a bottleneck, using EoT can therefore be useful. Otherwise, it is not worth the computational cost.

Table 6: $R@1$ for different transformations applied before search, with or without EoT when activating the images. Results are averaged on 10k images.

| | Activation | Identity | Contr. 0.5 | Contr. 2.0 | Bright. 0.5 | Bright. 2.0 | Hue 0.2 | Blur 2.0 | JPEG 50 | Rot. 25 | Rot. 90 | Crop 0.5 | Resi. 0.5 | Meme | Random | Avg. |
|---|---|---|---|---|---|---|---|---|---|---|---|---|---|---|---|---|
| Without EOT | 40 ms | 1.00 | 1.00 | 0.96 | 1.00 | 0.92 | 1.00 | 0.96 | 0.99 | 0.10 | 0.50 | 0.29 | 1.00 | 0.43 | 0.32 | 0.75 |
| With EOT | 870 ms | 1.00 | 1.00 | 0.95 | 1.00 | 0.92 | 1.00 | 0.95 | 0.99 | 0.14 | 0.64 | 0.33 | 1.00 | 0.45 | 0.33 | 0.76 |

# D    ACTIVE INDEXING VS. WATERMARKING

**Discussion.**    Watermarking and active indexing both modify images for tracing and authentication, however there are significant differences between them. Watermarking embeds arbitrary information into the image. The information can be a message, a copyright, a user ID, etc. In contrast, active indexing modifies it to improve the efficiency of the search engine. Watermarking also focuses on the control over the False Positive Rate of copyright detection, *i.e.* a bound on the probability that a random image has the same message as the watermarked one (up to a certain distance).

Although watermarking considers different settings than indexing methods, it could also be leveraged to facilitate the re-identification of near-duplicate images. In this supplemental section, we consider it to address a use-case similar to the one we address in this paper with our active indexing approach. In this scenario, the watermark encoder embeds binary identifiers into database images. The decoded identifier is then directly mapped to the image (as the index of a list of images).

**Experimental setup.**    In the rest of the section, we compare active indexing against recent watermarking techniques based on deep learning.

- For indexing, we use the same setting as in Sec. 5.1 (IVF-PQ index with 1M reference images). When searching for an image, we look up the closest neighbor with the help of the index.
- For watermarking, we encode 20-bit messages into images, which allows to represent $2^{10} \approx 10^6$ images (the number of reference images). When searching for an image, we use the watermark decoder to get back an identifier and the corresponding image in the database.

Like before, we use $R@1$ as evaluation metric. For indexing, it corresponds to the accuracy of the top-1 search result. For watermarking, the $R@1$ also corresponds to the word accuracy of the decoding, that is the proportion of images where the message is perfectly decoded. Indeed, with 20-bit encoding almost all messages have an associated image in the reference set, so an error on a single bit causes a mis-identification (there is no error correction[1]).

We use two state-of-the-art watermarking methods based on deep learning: SSL Watermarking (Fernandez et al., 2022), which also uses an adversarial-like optimization to embed messages, and HiDDeN (Zhu et al., 2018), which encodes and decodes messages thanks to Conv-BN-ReLU networks. The only difference with the original methods is that their perturbation $\delta$ is modulated by the hand-crafted perceptual attenuation model presented in App. A. This approximately gives the same image quality, thereby allowing for a direct comparison between active indexing and watermarking.

**Results.**    Tab. 7 compares the $R@1$ when different transformations are applied before search or decoding. Our active indexing method is overall the best by a large margin. For some transformations, watermarking methods are not as effective as passive indexing, yet for some others, like crops for HiDDeN, the watermarks are more robust.

Table 7: $R@1$ for different transformations applied before search, when using either watermarking or active indexing. Results are averaged on 1k images. Best result is in **bold** and second best in *italic*.

| | Identity | Contr. 0.5 | Contr. 2.0 | Bright. 0.5 | Bright. 2.0 | Hue 0.2 | Blur 2.0 | JPEG 50 | Rot. 25 | Rot. 90 | Crop 0.5 | Resi. 0.5 | Meme | Random | Avg. |
|---|---|---|---|---|---|---|---|---|---|---|---|---|---|---|---|
| Passive indexing | **1.00** | 0.73 | 0.39 | 0.73 | 0.28 | 0.62 | 0.48 | *0.72* | *0.07* | 0.14 | 0.14 | *0.72* | 0.14 | 0.13 | 0.45 |
| Active indexing (ours) | **1.00** | **1.00** | **0.96** | **1.00** | **0.92** | **1.00** | **0.96** | **0.99** | **0.10** | **0.50** | *0.29* | **1.00** | 0.43 | **0.32** | **0.75** |
| SSL Watermarking (Fernandez et al., 2022) | **1.00** | *0.98* | *0.53* | *0.98* | *0.63* | *0.85* | 0.13 | 0.00 | 0.00 | *0.15* | 0.11 | 0.00 | *0.46* | 0.07 | 0.42 |
| HiDDeN[2] (Zhu et al., 2018) | *0.94* | 0.87 | 0.36 | 0.85 | 0.55 | 0.00 | *0.81* | 0.00 | 0.00 | 0.00 | **0.92** | 0.44 | **0.77** | *0.16* | *0.48* |

---

[1] In order to provide error correction capabilities, one needs longer messages. This makes it more difficult to insert bits: in our experiments, with 64 bits we observe a drastic increase of the watermarking bit error rate.

[2] Our implementation. As reported in other papers from the literature, results of the original paper are hard to reproduce. Therefore to make it work better, our model is trained on higher resolution images ($224 \times 224$), with a payload of 20-bits, instead of 30 bits embedded into $128 \times 128$. Afterwards, the same network is used on images of arbitrary resolutions, to predict the image distortion which is later rescaled as in Eq. (8). In this setting the watermark can not always be inserted (6% failure).

# E   MORE QUALITATIVE RESULTS

Figure 10 gives more examples of activated images from the DISC dataset, using the same parameters as in Sec. 5.2. The perturbation is very hard to notice (if not invisible), even in flat areas of the images because the perceptual model focuses on textures. We also see that the perturbation forms a regular pattern. This is due to the image (bilinear) resize that happens before feature extraction.

Figure 9 gives example of an image activated at several values of perturbation strength $\alpha$ of Eq. (8) (for instance, for $\alpha = 20$ the image has PSNR 27dB and for $\alpha = 1$ the image has PSNR 49dB). The higher the $\alpha$, the more visible the perturbation induced by the activation is. Nevertheless, even with low PSNR values ($< 35$dB), it is hard to notice if an image is activated or not.

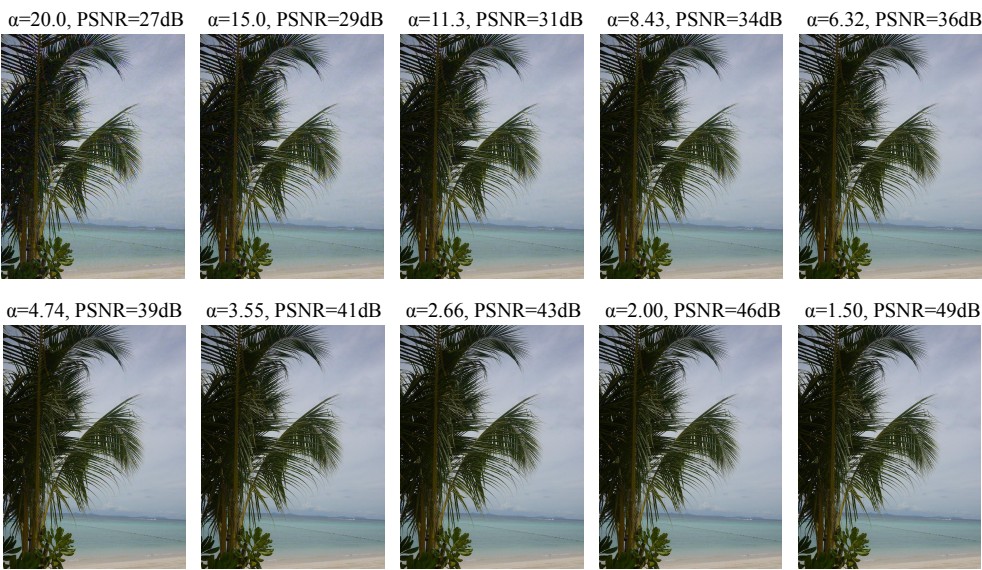

Figure 9: Example of one activated image at different levels of $\alpha$.

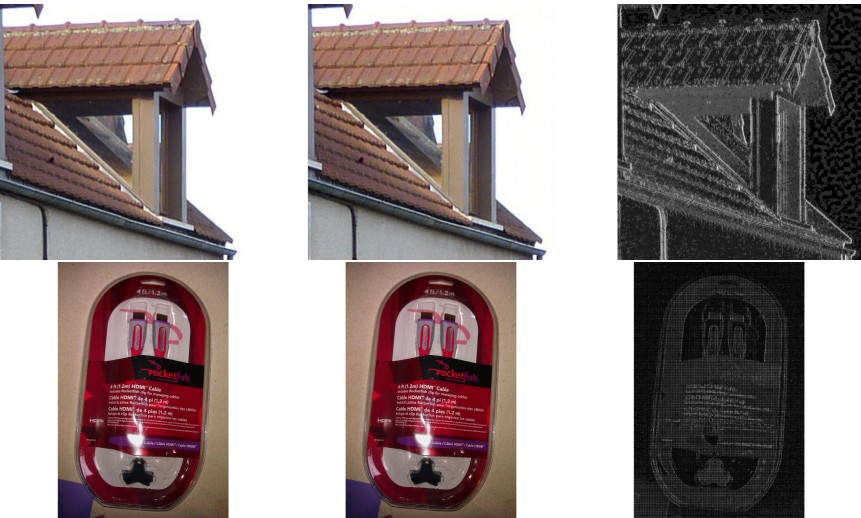

Figure 10: Example of activated images for $\alpha = 3.0$. (Left) original images, (Middle) activated images, (Right) pixel-wise difference.

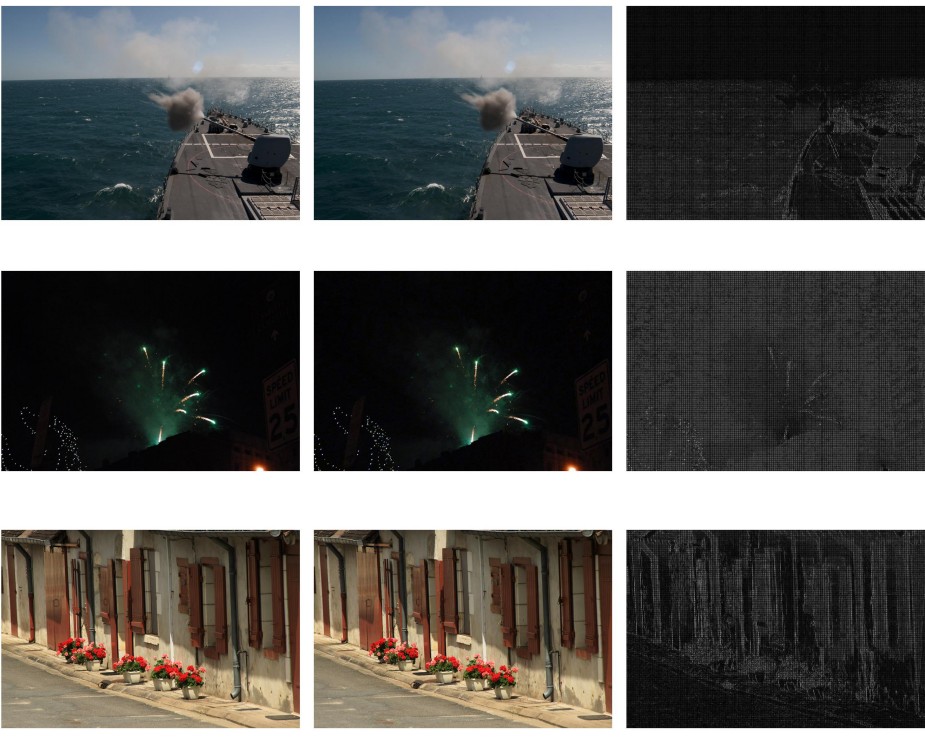

Figure 10: Example of activated images for $\alpha = 3.0$. (Left) original images, (Middle) activated images, (Right) pixel-wise difference. Images are R000005.jpg, R000045.jpg, R000076.jpg, R000172.jpg and R000396.jpg.

