# OpenReview forum: "Active Image Indexing"
_ICLR.cc/2023/Conference — ICLR 2023 poster_

### Official Review · Reviewer_pXd8 · 2022-10-25

**Confidence:** 4
**Correctness:** 4
**Technical Novelty And Significance:** 2
**Empirical Novelty And Significance:** 2
**Recommendation:** 6

**Clarity, Quality, Novelty And Reproducibility:**

- The paper is well written and easy to understand

- The application of this approach is constrained by being able to apply some transformation on the image and indexing it before releasing it to public. Also, how and who will hold that index and make it available for CID applications? In my opinion, this is a huge constrain specifically if applied to real world applications and constrains its usefulness to only big search or content sharing companies like Google or Facebook, or to be used in a blockchain type of application. I would like to see more explanations on how such a technique could be used to solve real world problems.

- What's the effect of adding new images to the index?
Specifically, this approach uses K-Means for space partitioning, which could be sensitive to how much the images used for training are representative of all images the system is able to index and retrieve, whether those are domain specific or not. In other words, an evaluation (or at least a discussion) showing that adding more images to the index (in any iterative way <> a one time indexing) will not impact the space partitioning and the copy retrieval.

**Strength And Weaknesses:**

Strengths of the paper:
- Active indexing outperforms passive indexing by applying some perturbations on the original image before publication
- Active indexing seems to generalize well with other feature extractors and other indexes

Areas where the paper could improve:
- The paper lacks some comparison to state of the art approaches for image copy detection
- The queries used in the evaluation are not the queries from DISC21 dataset since the authors needed to control the image transformations. This makes the obtained results not comparable to other work using the same dataset. In my opinion a fair comparison to other existing work is needed to validate this approach, specifically that the contributions are incremental.




**Summary Of The Paper:**

This paper proposes an active image indexing approach that allows image copy detection. The proposed solution applies some perturbations to the images during the indexing process which improves similarity search on large databases. Those perturbations are claimed to be invisible to the human eye. Evaluations on the DISC21 dataset showed that active indexing outperforms passive indexing and that the method generalizes to other feature extractors and other indexes.


**Summary Of The Review:**

I like the proposed approach in this paper and the quality of the writing. The contributions of this paper are incremental and the paper lacks some comparison to some state of the art approaches for image copy detection (needed from my perspective to add credibility to how effective active indexing is). I also have some concerns regarding how useful such an approach is for real world applications, and even for solving research related problems.

---

> ### Author Response · Authors · 2022-11-11
> **Response to Reviewer pXd8**
>
> We thank you for the time you spent reviewing the paper and for your meaningful comments.
>
> - *(Cons) “The paper lacks some comparison to state of the art approaches for image copy detection” (in Strength and weaknesses)*
>
> We are not aware of more recent or relevant publications on image copy detection than the open-source SSCD method used in this work, that is the state of the art in the field.
> We are open to suggestions for other comparisons that could be relevant.
>
> - *(Cons) “The queries used in the evaluation are not the queries from DISC21 [...]. In my opinion a fair comparison to other existing work is needed to validate this approach, specifically that the contributions are incremental.” (in Strength and weaknesses)*
>
> As we state in the paper, it is not possible to “replay” the original DISC21 transformations, as  “most transformations of DISC21 were done manually so they are not reproducible”. However, we did our best to sample the most relevant transformations from the original dataset.
>
> Besides, we believe the contributions are more than just incremental, since the concept of active indexing has not been explored before to the best of our knowledge.
>
> - *“The application of this approach is constrained by being able to apply some transformation on the image and indexing it before releasing it to public.
> Also, how and who will hold that index and make it available for CID applications? In my opinion, this is a huge constrain specifically if applied to real world applications and constrains its usefulness to only big search or content sharing companies like Google or Facebook, or to be used in a blockchain type of application. I would like to see more explanations on how such a technique could be used to solve real world problems. (in Clarity, Quality, Novelty And Reproducibility)*
>
> Active indexing could be used in several real world applications, not only by big companies like Google or Facebook. The method could apply for all situations where “Alice” distributes images and wants to moderate versions edited and shared by “Bob” over the web (and where doing a brute force search in the embeddings would be too time consuming).
> For example, stock photography providers (like Shutterstock, Getty)  want to check that their pictures are rightfully credited, or Digital Asset Management companies in general. They would just distribute images activated for the index and feature extractor they use.
> Another example can be with generative image models (DALL.E and Stable Diffusion) and APIs distributed by OpenAI (https://labs.openai.com) and HuggingFace (https://huggingface.co/spaces/runwayml/stable-diffusion-v1-5). These providers may want to monitor the presence of their images on the internet, to ensure that the generated images comply with the content policy.
>
> - *"What's the effect of adding new images to the index? Specifically, this approach uses K-Means for space partitioning, which could be sensitive to how much the images used for training are representative of all images the system is able to index and retrieve, whether those are domain specific or not. In other words, an evaluation (or at least a discussion) showing that adding more images to the index (in any iterative way <> a one time indexing) will not impact the space partitioning and the copy retrieval. (in Clarity, Quality, Novelty And Reproducibility)*
>
> In our case, the index is trained on the train set of DISC21, and the quantizer is trained at this moment. Once it is trained, adding an image simply updates the inverted-file structure to store the new vector, but it does not change the space partitioning, so adding an image has no effect on the index.
> That being said, our method is specifically designed for a trained version of the index.

---

### Official Review · Reviewer_DYH1 · 2022-10-27

**Confidence:** 3
**Correctness:** 3
**Technical Novelty And Significance:** 2
**Empirical Novelty And Significance:** 2
**Recommendation:** 6

**Clarity, Quality, Novelty And Reproducibility:**

The results are not easily reproducible. The assumptions the paper makes are several. Nevertheless, the methodology is described sufficiently, and the novelty is considered satisfactory.

**Strength And Weaknesses:**

Pros:
•	The paper is well-written and coherent, and the methods' explanation is illustrative.
•	The experimental results showcase that the proposed method significantly reduces the time complexity of searching similar images without compromising the retrieval accuracy.
•	Active indexing can be used selectively in the database to enhance retrieval performance on important images without affecting the performance of passive indexing.
Cons:
•	This method assumes that the images can be processed before extracting the descriptors and defining the indexing structure, and consequently, the system cannot be used as a black box.
•	Although the search speed is reduced considerably, the retrieval performance results are inconclusive. For example, in the case of IVF 512, PQ 32X8, the performance boost of active indexing is negligible.
•	The neural network training time for active indexing is time-consuming.

Minor details:
•	Different symbols could be used in equation (4), as the symbol x can be confusing because it refers to both the originally extracted descriptor and in the new split vector.


**Summary Of The Paper:**

This paper addresses the issue of image copy detection and retrieval from large databases by incorporating a robust feature extractor and a scalable, efficient search algorithm. The feature extractor is a neural network that has been trained to extract descriptors that are robust to various transformations. The search algorithm is based on the Inverted Files Method followed by a Product Quantization technique to compress the images’ descriptors and significantly reduce retrieval time. Experiments are provided to demonstrate the applicability of the method.

**Summary Of The Review:**

The paper marginally passes the acceptance threshold since the method assumes that the images can be processed before extracting the descriptors and defining the indexing structure. Consequently, the system cannot be used as a black box. Although the search speed is reduced considerably, the retrieval performance results are inconclusive. For example, in the case of IVF 512, PQ 32X8, the performance boost of active indexing is negligible.

---

> ### Author Response · Authors · 2022-11-11
> **Response to Reviewer DYH1**
>
> Thank you for the feedback.
>
> - *"This paper addresses the issue of image copy detection and retrieval from large databases by incorporating a robust feature extractor and a scalable, efficient search algorithm. [...]. Experiments are provided to demonstrate the applicability of the method." (in Summary of the paper)*
>
> Your summary does not mention the main contribution, which is to make image indexing more robust by pro-actively modifying images before their release.
>
> - *(Pro) " The experimental results showcase that the proposed method significantly reduces the time complexity of searching similar images without compromising the retrieval accuracy." (in Strength and weaknesses)*
>
> To be more precise we improve the accuracy of image indexing. For indexing methods like IVFPQ that offer a tradeoff between accuracy and time complexity, our technique can either  improve the accuracy for the same speed or, as you say, for a given retrieval accuracy budget, reduce the time complexity without compromising the retrieval accuracy.
>
> - *(Con) "This method assumes that the images can be processed before extracting the descriptors and defining the indexing structure"
> (Con) "The neural network training time for active indexing is time-consuming." (in Strength and weaknesses)*
>
> Our method does not assume that the images can be modified before defining and updating the index. Indeed, the original and activated images are indexed exactly the same way. Moreover the method intervenes on a pre-trained neural network for embedding extraction, so there might be a misunderstanding here.
>
> - *(Con) “the system cannot be used as a black box.” (in Strength and weaknesses)*
>
> We agree that the system can not be used as black box: the image owner has  to know the indexing system and the feature extractor that are going to be used. However it is not a strong limitation here, since the application scenario is that the owner of the images also builds the index to trace their diffusion.
>
> - *(Con) "Although the search speed is reduced considerably, the retrieval performance results are inconclusive. For example, in the case of IVF 512, PQ 32X8, the performance boost of active indexing is negligible." (in Strength and weaknesses)*
>
> As we point out in the paper, “the impact [of active indexing] is smaller when the indexing method is more accurate”, i.e. the method works better at low-time complexity operating points.  The specific index structure IVF 512, PQ 32X8 (+2% gain in avg. R@1) is not representative of the general results (e.g. IVF4096,PQ8x8: +30% - PCA64 LSH: +11% - IVF flat 1024: +36%).
>
> - *"Different symbols could be used in equation (4), as the symbol x can be confusing because it refers to both the originally extracted descriptor and in the new split vector." (in Strength and weaknesses)*
>
> Thank you for the suggestion, we have consequently changed $x^1$ into $u^1$, in the equation (4).
>
> Please let us know if you have any further comments.

---

### Official Review · Reviewer_gK1Y · 2022-10-31

**Confidence:** 4
**Correctness:** 4
**Technical Novelty And Significance:** 3
**Empirical Novelty And Significance:** 3
**Recommendation:** 8

**Clarity, Quality, Novelty And Reproducibility:**

The paper is very well written and easy to follow and the idea of active indexing is novel. The authors promised to release the implementation, so the experiments should be reproducible.

**Strength And Weaknesses:**

Pros:
- The paper is very well written and very well motivated by pointing out the challenges that indexing faces regarding copy detection and proposing a solution for it.
- I really like the main idea proposed in this paper of imperceptibly modifying the image in order to push its represnation to the center of the indexing partition to make it more indexing friendly.
- The paper contains a thorough experimental analaysis that justifies the authors claims and shows that their method significantly improves retrieval performance on a number of neural extractors and indexing structures


Cons:
- The fact that the image has to be modified before its release is a big limitation since it cannot be applied to already existing databases where images have already been released. However, it can still be applied to new images released in the same database, benefiting all future releases.

**Summary Of The Paper:**

This paper addresses the problem of image copy detection through image retrieval and proposes a method that optimizes jointly both representation learning and approximate similarity search. Large databases rely on indexes for approximate and efficient search, but the challenge comes when copies are significantly modified and then its representation switches indexing partitions and the closest neighbour cannot be returned. The authors address this problem by modifying the image in a way that its representation is pushed towards the center of the indexing partition, giving more "room" for edited images to still fall in the same partition as the original copy. This is done by minimizing and indextation loss back to the image pixels, assuming that the image can be modified before its release.

**Summary Of The Review:**

The paper is well written, the idea is both interesting and novel, and the experimental analysis is extensive and thorough. The method has a few limitations (not only the adversial attacks discussed by the authors but the fact that images need to be activated before its release) but they do not have a large impact on my positive opinion towards the paper.

---

> ### Author Response · Authors · 2022-11-11
> **Response to Reviewer gK1Y**
>
> We thank you for your extensive read of the paper, the quality of your comments and your overall positive opinion.
>
> - *(Con) "The fact that the image has to be modified before its release is a big limitation since it cannot be applied to already existing databases where images have already been released. However, it can still be applied to new images released in the same database, benefiting all future releases." (in Strength And Weaknesses)*
>
> We agree with this statement. What can be done however, is to change the image for future releases. Conceivably there could  be 2 versions of the image online, a passive one that can be retrieved as long as it is not transformed too strongly, and an activated one which has better copy detection properties.
> Note that no change is needed to the index as both images share the same indexed descriptor.
>
> - *"[...] The method has a few limitations (not only the adversial attacks discussed by the authors but the fact that images need to be activated before its release) but they do not have a large impact on my positive opinion towards the paper." (in Summary of the Review)*
>
> Thank you for the feedback. We have updated the limitations in our conclusion, according to your comments.

---

> > ### Comment · Reviewer_gK1Y · 2022-11-18
> > **Response to the authors**
> >
> > Thank you for your response to my comments.
> >
> > *What can be done however, is to change the image for future releases. Conceivably there could be 2 versions of the image online, a passive one that can be retrieved as long as it is not transformed too strongly, and an activated one which has better copy detection properties. Note that no change is needed to the index as both images share the same indexed descriptor.*
> >
> > I did not have into account that both images share the same indexed descriptor. That is actually a very nice feature of the method.

---

### Decision · Program_Chairs · 2023-01-20

**Decision:**

Accept: poster

**Justification For Why Not Higher Score:**

the solution proposed will be more relevant to image indexing and retrieval researchers within the broader ICLR researchers

**Justification For Why Not Lower Score:**

the paper has sufficient merits in novelty and results, and it is well written

**Metareview: Summary, Strengths And Weaknesses:**

This paper aims to solve problem with image copy detection through image retrieval from large databases by incorporating a robust feature extractor and a scalable, efficient search algorithm. The proposed solution involves modifying the image in a way that its representation is pushed towards the center of the indexing partition, giving more "room" for edited images to still fall in the same partition as the original copy. The idea is rather interesting and novel, and the experimental results and analysis are good.

**Note From Pc:**

if the above contains the word "oral" or "spotlight" please see: "oral" presentation means -> notable-top-5% and "spotlight" means -> notable-top-25%. As stated in our emails, we are disassociating presentation type from AC recommendations

**Summary Of Ac-Reviewer Meeting:**

NA as all reviewers accept the paper.